# Identification of Flavone Derivative Displaying a 4′-Aminophenoxy Moiety as Potential Selective Anticancer Agent in NSCLC Tumor Cells

**DOI:** 10.3390/molecules28073239

**Published:** 2023-04-05

**Authors:** Giovanna Mobbili, Brenda Romaldi, Giulia Sabbatini, Adolfo Amici, Massimo Marcaccio, Roberta Galeazzi, Emiliano Laudadio, Tatiana Armeni, Cristina Minnelli

**Affiliations:** 1Department of Life and Environmental Sciences, Marche Polytechnic University, 60131 Ancona, Italy; g.mobbili@univpm.it (G.M.); giulia.sabbatini@staff.univpm.it (G.S.); r.galeazzi@univpm.it (R.G.); 2Department of Specialist Clinical Sciences, School of Medicine, Marche Polytechnic University, 60131 Ancona, Italy; b.romaldi@pm.univpm.it (B.R.); a.amici@staff.univpm.it (A.A.); t.armeni@univpm.it (T.A.); 3Department of Chemistry G. Ciamician, University of Bologna, Via Selmi 2, 40126 Bologna, Italy; massimo.marcaccio@unibo.it; 4Department of Science and Engineering of Matter, Environment and Urban Planning, Marche Polytechnic University, 60131 Ancona, Italy; e.laudadio@staff.univpm.it

**Keywords:** flavone synthesis, non-small cell lung cancer, phenoxy groups, selective anticancer drugs

## Abstract

Five heterocyclic derivatives were synthesized by functionalization of a flavone nucleus with an aminophenoxy moiety. Their cytotoxicity was investigated in vitro in two models of human non-small cell lung cancer (NSCLC) cells (A549 and NCI-H1975) by using MTT assay and the results compared to those obtained in healthy fibroblasts as a non-malignant cell model. One of the aminophenoxy flavone derivatives (**APF-1**) was found to be effective at low micromolar concentrations in both lung cancer cell lines with a higher selective index (SI). Flow cytometric analyses showed that **APF-1** induced apoptosis and cell cycle arrest in the G2/M phase through the up-regulation of p21 expression. Therefore, the aminophenoxy flavone-based compounds may be promising cancer-selective agents and could serve as a base for further research into the design of flavone-based anticancer drugs.

## 1. Introduction

Various chemotherapeutic agents have been commonly and successfully used for the treatment of cancer [1,2,3]; however, the development of inducible drug resistance and the difficulty to limit the associated side effects are often the main cause of chemotherapy failure. Therefore, the identification of novel anticancer drugs with great cancer-selective toxicity remains the main challenge for medicinal chemistry researchers [4,5,6,7]. The flavone scaffold has emerged as a versatile source in the field of drug discovery (Figure 1A). It is widely used as a building block for design and synthesis of novel lead compounds that exhibit anticancer activity against a wide range of human cancerous cell lines [8]. Between them, amino-functionalized flavone derivatives have been shown to have high selectivity towards some cancer types and are considered as a new promising class of anti-cancer agents. The most popular example is the AF or [(5-amino-2,3-fluorophenyl)-6,8-difluoro-7-methyl-4H-1-benzopyran-4-one, NSC 686288)] in which two amino groups are directly linked to the aromatic sp^2^ carbon of the flavonoid ring system (Figure 1). Its selective antiproliferative activity has been demonstrated in vitro and in vivo in human breast and renal cancer cell lines [9,10]. The compound AFP646, a lysine prodrug of AF, has been investigated in phase-1 study and appears to be useful against solid tumors. A sub-class of the amino-functionalized flavone derivatives (Figure 1B) could include compounds containing tertiary ammines such as Flavopiridol (Alvocidib, L86-8275, NSC 649890, HMR 1275), an antineoplastic agent that inhibits cyclin-dependent kinases (CDKs) [11], and LW-213, which induces G2/M cell cycle arrest in breast cancers cells by suppression of Akt/Gsk3b/β-catenin or reducing Cyclin/B1/CDC2 complex in chronic myeloid leukemia [12,13].

Among the anticancer drugs approved by the FDA, there are compounds bearing phenoxy moiety which seems to be crucial for their high cancer antiproliferative activity [14]. Interestingly, the anticancer potential of the flavone derivatives in which the amino group is linked to the scaffold through a phenoxy moiety has never been reported. Therefore, the present study aims to ascertain the potential of this class of flavone derivatives as a chemical scaffold or lead compound for the synthesis of novel anticancer entities. In this context, starting from our previous work, in which we designed several nitroflavone-based compounds [15], we synthetized the corresponding 4-aminophenoxy flavone derivatives (**APF**) by reduction of the aromatic nitro groups to amines. Therefore, we explored how the position of the 4-aminophenoxy linked to the A ring of flavone scaffold as well as the presence of the phenoxy moiety on the B ring influences the selective cytotoxicity of the compounds against non-small lung cancer (NSCLC) cells. From the results obtained, a hit compound able to selectively induce apoptosis and G2/M cell cycle arrest in NSCLC was identified. These findings can stimulate further development of aminophenoxy flavone derivatives as novel anticancer agents.

## 2. Results and Discussion

### 2.1. Chemistry

With the aim to investigate the influence of the 4-aminophenoxy position on the A ring, five aminophenoxy flavone derivatives were synthetized. All the **APF** compounds were prepared using the synthetic route shown in Figure 1.

The starting materials 2,5-dihydroxyacetophenone (1a), 2,4-dihydroxyacetophenone (1b), and 2,6-dihydroxyacetophenone have been O-arylated with 4-fluoronitrobenzene in the presence of Cs_2_(CO)_3_ to obtain 2-hydroxy-5-(4-nitrophenoxy)acetophenone (2a), 2-hydroxy-4-(4-nitrophenoxy)acetophenone (2b) and 2-hydroxy-6-(4-nitrophenoxy)acetophenone (2c), respectively. The following crossed aldol condensation with the appropriate aldehyde led to the correspondent chalcones, which were oxidized to nitrophenoxy flavone derivatives in the presence of DMSO-I_2_ (3a–e). Then, the use of Ni(acac)_2_ and PMHS allowed for selectively reducing the nitro group and obtaining the **APF-1** [11], **APF-2**, **APF-3**, **APF-4**, and **APF-5**. The overall structure of APF and the related yields are shown in the Figure 1.

The cytotoxicity of **APF** derivatives could be influenced by their ability to cross the cellular membrane and reach their potential targets. Therefore, to assess the cell permeability, we experimentally determined the n-octanol-water partition coefficient (K_o/w_) and the Lipinski’s Rule violations using SwissADME software. As shown in the Table 1, there were no significative differences in the calculated LogP value for **APF** containing the phenoxy group linked to the B ring of flavone scaffold. As expected, the **APF-2** which has only the 4-aminophenoxy moieties linked to the A ring, showed a lower LogP (2.3 vs. 2.9). Note the overall acceptable to good “druggability profile” of **APF**, with no violations of the Lipinski’s Rule for four indicators (molecular weight, LogP, hydrogen bond donor and acceptor groups).

### 2.2. Influence of 4-Aminophenoxy and Phenoxy Moieties on Cancer Cytotoxicity of Flavone-Based Derivatives

To investigate the potential of aminophenoxy flavone derivatives as selective anticancer drugs, the half maximal inhibitory concentration (IC50), and therefore the cytotoxicity of **APF** derivatives, was assessed by MTT assay on NSCLC cells (H1975 and A549) exposed for 72 h to increasing concentrations of the synthetized compounds (0–160 µM) (Appendix A; Table 1). The results obtained were compared with those achieved on equally treated human dermal fibroblasts (HDF) which are usually used as normal cell controls in almost all screening studies [16,17,18]. For each compound tested we calculated the selective index (SI) which can be defined as the ratio of the cytotoxic concentration of compound against its effective bioactive concentration [19]. The evaluation of the SI value for any anti-cancer compound is very crucial for determining its potential safety [20,21]. Overall, a low SI (<1) means that the compound is toxic, while an SI ≥ 10 is recommended as minimal value for a potential selective drug that can be further investigated [20,22]. As shown in Table 2, **APFs** showed cytotoxicity in both NSCLC cell lines, although with different efficacy and selectivity.

In the presence of a 4-aminophenoxy group at the 6 position of the A ring and a terminal phenoxy group on the B ring (**APF-1** and **APF-5**), the compounds showed better cancer-selective cytotoxicity. Between them, the **APF-1**, containing a phenoxy moiety linked to C’3 position of the B ring, was the most effective with an IC50 at low micromolar concentration of 4 and 2 µM for A549 and H1975, respectively and an SI > 10. Interestingly, the absence of the phenoxy moiety linked to the B ring (**APF-2**) and the different position of the 4-aminophenoxy moieties on the A ring (**APF-3**, **APF-4**) decreased the efficacy and selectivity of aminophenoxy derivatives compared to **APF-1** (Figure 2).

Treatment of advanced NSCLC with the anthracycline antibiotic doxorubicin, provides an overall response rate of only 30–50% [23,24,25]. Moreover, its acute and cumulative dose-related toxicity limited its therapeutic potential [26,27]. As shown in Table 2, **APF-1** provided a higher SI in H1975 (~4.5-fold increase) while in the A549, the SI was similar to those obtained with our compound. According to the LogP values, the higher cytotoxic effect of **APF-1** was not associated to a different lipophilicity, but could be due to its ability to specifically interact with cellular target overexpressed and/or overactivated in cancer cells. Therefore, considering the cytotoxicity and the selectivity index results of the synthesized derivatives, we further evaluated the effect on tumor cells of compound **APF-1**.

### 2.3. **APF-1** Selectively Induced Cell Apoptosis in NSCLC Cells

Annexin V-FITC and propidium iodide staining is a widely used method for the quantification of cellular apoptosis and necrosis. In the presence of calcium ions, Annexin V specifically binds to phosphatidylserine (PS), a membrane phospholipid which is exposed at the outer plasma membrane leaflet during early and late apoptosis [28]. Propidium iodide, a phenanthrene-based dye, is a non-specific DNA intercalating agent which can only enter cells with membrane damage [29]. In order to detect the early stages of apoptosis, cells were treated for 24 h with an **APF-1** concentration near the IC50 of A549 and H1975 (1.5 and 3 µM). Flow cytometric apoptosis analysis of NSCLC and HDF cell lines are presented in Figure 3. The untreated H1975 cells showed about 84% live cells, while we observed a significant increase in cell death at the lower **APF-1** concentration (1.5 µM) with 16% of early apoptotic cells and 11% of late apoptotic cells. The concentration of 3 µM caused an even greater increase in late apoptosis and the percentage of live cells was 46% (Figure 3A). Here, we observed 2.6- and 6.5-fold increase, respectively, of early and late apoptosis, compared to untreated cells. In the untreated A549 cells, the percentage of live cells was about 93%, which reached 73% after treatment with 3 µM **APF-1**. At this concentration, we observed a marked increase in the late apoptotic cells from 3.4% of untreated cells to 20% (5.9-fold increase, *p* < 0.01); a 2-fold increase in early apoptosis with respect to untreated cells was also observed (Figure 3B).

In the untreated HDF, the live cells reached about 94% with only a slight decrease after **APF-1** treatment (1.5 µM, 92% and 3 µM, 86%). We did not detect any effect on the number of the late apoptotic cells, while an increase in the early apoptosis was observed only at the highest **APF-1** concentration (4.7-fold-increase, *p* < 0.01) (Figure 3C). However, as shown in Figure 3D, the induction of apoptosis in cancer cells, defined as the sum of early and late apoptosis, was about 4 times higher with respect to untreated cells, while in fibroblasts it was lower. Overall, **APF-1** induced cell death mainly by apoptosis induction and the effect was higher toward cancer cells with respect to normal fibroblasts.

### 2.4. **APF-1** Induces G2/M Cell Cycle Arrest in NSCLC Cells

To further investigate the inhibitory effects of **APF-1** on the NSCLC cell proliferation, we examined by flow cytometry after labeling with propidium iodide, the cell cycle phases distribution 24 h after **APF-1** treatment (Figure 4). The results were compared with those obtained in the HDF cells. The proportion of cells in the G0/G1, S, and G2/M phases of the cell cycle was determined using the Multicycle function in FCS Express 7 Software (DeNovo Software, Pasadena, CA, USA). Overall, **APF-1** affected the cell cycle distribution in a dose-dependent manner in both NSCLC cell lines (H1975 and A549) by inducing a marked increase in the G2/M phase population; the concomitant decrease in the percentage of cells in G0/G1 phase suggests that G2/M phase arrest was induced by **APF-1** [30]. Specifically, in the H1975 cells treated with 3 µM **APF-1**, we observed an increase in the population in the G2/M phase from 36 to 70% compared to untreated cells (2-fold increase, * *p* < 0.01), while the percentages of cells in the S and G0/G1 phase are significantly decreased (13% vs. 5.6% and 50% vs. 24%, respectively). In A549 cells treated with 3 µM **APF-1**, we observed a significant increase in the percentage of cells in the G2/M phase compared to untreated control cells (75% vs. 18%). Treatment with 1.5 µM **APF-1** also induced significant changes in the percentages of cells in the different phases of the cell cycle, again with a cell cycle arrest in the G2/M phase (2.8-fold increase compared to the untreated cells). The untreated HDF cell line showed a typical cell cycle profile of normal fibroblasts [31,32] and interestingly, it was not affected by the **APF-1** treatment, thus indicating the ability of **APF-1** to selectively induce cell cycle arrest in the G2/M phase only in NCSLCs.

To further investigate the pathways affected by **APF-1**, we examined the expression level of p21^waf1/cip1^ protein. p21^waf1/cip1^ protein, primarily a cyclin-dependent kinase (CDK) inhibitor, plays a key role in the regulation of proliferation through the control of the cell cycle progression, induction of apoptosis, and transcriptional regulation [33]. Indeed, up-regulated p21^waf1/cip1^ promotes cell cycle arrest by broadly binding with several cyclin/CDK complexes and consequently inhibiting their kinase activity [33,34,35]. As a proliferation inhibitor, p21^waf1/cip1^ plays an important role in controlling tumor development and a decreased expression of p21^waf1/cip1^ is associated with a poor prognosis in lung cancer [36]. As shown in Figure 4D,E, **APF-1** treatment induced an increase in the p21^waf1/cip1^ protein level in both NSCLC cells (H1975 and A549) although with different efficacy. In the A549 cells, we observed an increase in the p21^waf1/cip1^ expression at the higher **APF-1** concentration (1.5-fold increase with respect to untreated cells, *p* < 0.5) while in the H1975 cells, a 3-fold increase in the p21 level with respect to untreated cells (*p* < 0.05) both at 1.5 and 3 µM **APF-1** concentrations was observed. However, in the HDF cell line we did not observe any significant change in the p21 expression, highlighting the selective activity of **APF-1**. It is known that the up-regulation of p21 could also result in the inhibition of G2/M phase transition [33,37,38,39]. Therefore, based on these data, **APF-1** seems to trigger G2/M-phase arrest via p21 activation selectively in cancer cell lines.

## 3. Materials and Methods

### 3.1. Materials and Reagents

The materials and reagents used in the synthetic procedures were purchased from Sigma Aldrich Co. (Stenheim, Germany) and used without purification. All solvents were analytically pure and dried before use. TLC was carried out on aluminum sheets precoated with silica gel 60 F254 (Merck). Column chromatography was performed using silica gel 60 (230–400 mesh).

High-resolution MS (HRMS) ESI analyses were performed on a Xevo G2-XSQTof (Waters) mass spectrometer. Mass spectrometric detection was performed in the in the positive ion mode. The ^1^H and ^13^C NMR spectra were recorded at 400 and 100 MHz, respectively, on an Agilent Technologies 400 MHz Premium Shielded spectrometer. Chemical shifts (δ) are reported in ppm relative to TMS and coupling constants (J) in Hz. Cell culture reagents were obtained from Euroclone (Milan, Italy). Chemical reagents and propidium iodide (PI) were obtained from Sigma Aldrich (St. Louis, MO, USA). The Annexin V-FITC apoptosis detection kit was obtained from Biolegend (San Diego, CA, USA). Chemiluminescent substrate and secondary antibody (32460) were obtained from Thermo Fisher Scientific (Waltham, MA, USA). Bradford reagent and polyvinylidene difluoride (PVDF) membranes were obtained from Bio-Rad (Hercules, CA, USA).

### 3.2. Synthetic Procedure

The aminophenoxy flavone (**APF**) derivatives were obtained by reduction of the corresponding nitrophenoxy flavones, synthesized as previously described [11]. Ni(acac)_2_ (0.03 g, 0.1 mmol) and Poly(methylhydrosiloxane) (PMHS) (3 mL, 1.5 mmol) were added to a solution in dioxane (5 mL) of the nitrophenoxy flavone (1 mmol). The mixture was stirred at 80 °C under air atmosphere for 12 h. After completion of the reaction, monitored by thin-layer chromatography (silica gel), the mixture was cooled to room temperature and purified by silica gel column chromatography using ethyl acetate/cyclohexane as the eluent to obtain the desired compound. All compounds were characterized by NMR and HRMS (Appendix A).

6-(4-aminophenoxy)-2-(phenyl)-4H-chromen-4-one (APF-2) light yellow solid (0.21 g, Yield 65%). ^1^H NMR (DMSO-d6, 400 MHz): δ 5.09 (broad s, 2H, NH_2_), 6.62–6.67 (m, 2H), 6.82–6.88 (m, 2H), 7.00 (s, 1H), 7.22 (d, *J =* 3.1, 1H), 7.48 (dd, *J =* 3.1, *J =* 9.0, 1H), 7.54–7.64 (m, 3H) 7.79 (d, *J =* 9.0, 1H), 8.06–8.12 (m, 2H).^13^C NMR (DMSO-d6, 100 MHz): δ 106.1, 108.2, 114.9, 120.4, 121.3, 124.0, 124.0, 126.3, 129.1, 131.1, 131.8, 144.8, 146.1, 150.8, 156.7, 162.4, 176.7. Calcd. neutral mass for C_21_H_15_NO_3_: 329. 34,723 Da. HRMS: *m*/*z =* 330.32 [M+H]^+^, 352.37 [M+Na]^+^.

7-(4-aminophenoxy)-2-(3′-phenoxy)phenyl-4H-chromen-4-one (APF-3) light orange solid (0.23 g, Yield 55%). ^1^H NMR (DMSO-d6, 500 MHz): δ 5.14 (s, 2H, NH_2_), 6.61–6.67 (m, 2H), 6.84–6.89 (m, 2H), 6.95 (s, 1H), 6.97 (d, *J =* 2.44, 1H), 7.02 (d, *J =* 2.44, 1H), 7.03–7.06 (m, 2H), 7.13 (dd, *J =* 2.44, *J =* 8.24, 1H), 7.15–7.19 (m, 1H), 7.37–7.42 (m, 2H), 7.49–7.53 (m, 1H), 7.71–7.73 (m, 1H), 7.80–7.84 (m, 1H), 7.96 (d, *J =* 8.9, 1 H). ^13^C NMR (DMSO-d6, 125 MHz): δ 104.0, 107.8, 115.4, 115.6, 117.0, 118.4, 119.3, 122.0, 122.0, 124.3, 127.3, 130.6, 131.3, 133.6, 144.3, 147.1, 156.8, 157.6, 157.7, 162.0, 164.4, 176.8. Calcd. neutral mass for C_27_H_19_NO_4_: 421.131409 Da. HRMS: *m*/*z =* 422.14 [M+H]^+^, 444.12 [M+Na]^+^.

5-(4-aminophenoxy)-2-(3′-phenoxy)phenyl-4H-chromen-4-one (APF-4) yellow-orange powder (0.11 g, Yield 25%).^1^H NMR (DMSO-d6, 400 MHz): δ 5.04 (broad s, 2H, NH_2_), 6.59 (d, *J =* 8.2, 1H), 6.62 (d, *J =* 8.6, 2H), 6.78 (d, *J =* 8.6, 2H), 6.93 (s, 1H), 7.09 (d, *J =* 7.8, 2H), 7.17–7.22 (m, 2H), 7.34 (d, *J =* 7.8, 1H), 7.41–7.47 (m, 2H), 7.55–7.62 (m, 2H), 7.75–7.77 (m, 1H), 7.87 (d, *J =* 8.2, 1H). ^13^C NMR (DMSO-d6, 100 MHz): δ 108.7, 111.3, 111.8, 114.6, 114.9, 116.3, 118.7, 120.8, 121.3, 121.4, 123.8, 130.1, 130.8, 132.9, 134.0, 145.2, 145.7, 156.3, 157.2, 157.4, 158.4, 159.5, 176.2. Calcd. neutral mass for C_27_H_19_NO_4_: 421.131409 Da. HRMS: *m*/*z =* 422.14 [M+H]^+^, 444.12 [M+Na]^+^.

6-(4-aminophenoxy)-2-(4′-phenoxy)phenyl-4H-chromen-4-one (APF-5) yellow-orange powder (0.32 g, Yield 77%). ^1^H NMR (DMSO-d6, 500 MHz): δ 5.09 (broad s, 2H, NH_2_), 6.62–6.66 (m, 2H), 6.83–6.87 (m, 2H), 6.94 (s, 1H), 7.10–7.16 (m, 4H), 7.21 (d, *J =* 3.0, 1H), 7.23–7.26 (m, 1H), 7.44–7.49 (m, 3H), 7.77 (d, *J =* 8.9, 1H), 8.09–8.13 (m, 2H). ^13^C NMR (DMSO-d6, 125 MHz): δ 105.9, 108.8, 115.5, 118.5, 120.2, 120.8, 121.8, 124.4, 125.1, 126.2, 129.0, 130.8, 145.4, 146.6, 151.2, 155.8, 157.1, 160.6, 162.6, 170.8, 177.0. Calcd. neutral mass for C_27_H_19_NO_4_: 421.131409 Da HRMS: *m*/*z =* 422.14 [M+H]^+^, 444.12 [M+Na]^+^.

### 3.3. Log P Determination and Lipinski’s Rule Violations

LogP was determined as previously described [40]. Briefly, solutions of each **APF** were prepared in 1-octanol, and UV-vis spectra were acquired to determine the λ_max_. After having placed 2.5 mL of each solution in 10 mL conical centrifuge tubes, PBS was added (2.5 mL, pH 7.4) and the biphasic system was mixed using a vortex mixer (VELP Scientifica, Usmate Velate MB, Italy) followed by centrifugation (2000× *g*, 5 min). Then, the octanol phase absorption was measured at λ_max_. The partition coefficient K_o/w_ was calculated according to the following equation:K_o/w_ = [APF]_oct_/[APF]_PBS_ = A_f_/A_i_ × A_f_(1)
where A_i_ is the initial measured absorbance in octanol, pre-extraction, and A_f_ is the final measured absorbance in octanol, post-extraction.

Lipinski’s Rule violations were determined by using the free web service SwissADME. Available online: http://www.swissadme.ch (accessed on 10 February 2023).

### 3.4. Cell Culture

The human non-small cell lung cancer (NSCLC) and Human Dermal Fibroblasts (HDF) cell lines were all obtained from the American Type Culture Collection (ATCC). The H1975 (CRL-5908™) cell line was grown in RPMI 1640 medium (ECB2000) while the A549 (CCL-185™) and HDF were grown in complete DMEM medium (ECM0095) All cultures were supplemented with 10% fetal bovine serum (FBS), 2 mM L-glutamine, 100 U/mL penicillin, and 100 µg/mL streptomycin, while the A549 (CCL-185™) and HDF were grown in complete DMEM medium (ECM0095). All cell lines were routinely maintained in 75 cm^2^ flasks in a cell incubator at 37 °C, 5% CO_2_, and 95% relative humidity. The cell cultures were detached by trypsinization with 0.5% trypsin in PBS containing 0.025% EDTA. All cell culture reagents were supplied by Euroclone (Milan, Italy).

### 3.5. Cell Viability

The number of metabolically active cells, and thus cell viability, was assessed by a 3-(4,5-dimethylthiazol-2-yl)-2,5-diphenyltetrazolium bromide (MTT) assay as previously described [41]. The NSCLC cell lines (A549 and H1975) and HDF were seeded in 96-well plates to reach 50% confluence at 24 h; then, the medium was removed and replaced with 0.2 mL of fresh culture medium containing increasing compound concentrations (0–160 µM). After 72 h, the medium from each well was removed and replaced with fresh medium supplemented with MTT at a final concentration of 200 µg/mL. After incubation for 4 h at 37 °C, 0.2 mL of DMSO was added to each well followed by multiwell shaking for 4 min. The absorbance was read on a multiwell scanning microplate reader (BioTek Synergy HT MicroPlate Reader Spectrophotometer, BioTek Instruments Inc., Winooski, VT, USA) at 570 nm using the extraction buffer as a blank. The optical density in the control group (untreated cells) was considered as 100% viability. The relative cell viability (%) was calculated as (OD_570_ of treated samples/OD_570_ of untreated samples) × 100. Dose-dependent curves were therefore generated for the cytotoxic studies. The 50% inhibiting concentration (IC50) was determined by non-linear regression analysis with a three-parameter fit by utilizing SigmaPlot 12.0 Software, USA, Systat Software, Inc. SigmaPlot for Windows. Each experiment was performed at least five times in triplicate.

### 3.6. Apoptosis Evaluation

Apoptosis was analyzed by flow cytometry using an Annexin V-FITC apoptosis detection kit (Biolegend, San Diego, CA, USA), according to the manufacturer’s instructions. A549, H1975, and HDF cells were treated with three different concentrations of **APF-1** (1.5 and 3 µM). The analysis was performed after 72 h after treatment. Briefly, control and treated cells were trypsinized, washed twice with ice-cold PBS, and resuspended in 1X Annexin binding buffer at a final concentration of 1.0 × 106 cell/mL. A total of 5 µL Annexin V-FITC and 10 µL Propidium Iodide (PI) were added to the cell suspension and the mixture was incubated for 15 min at rt in the dark. Samples were analyzed using the Guava EasyCyte flow cytometer (Merk-Millipore, Louis Missouri, USA). A total of 5000 events were acquired for each sample. Annexin V-FITC was detected as a green fluorescence (λ_ex_ = 488 nm; λ_em_ = 516 nm) and PI was detected as a red fluorescence (λ_ex_ = 535 nm; λ_em_ = 617 nm). Early apoptosis is defined by Annexin V+/PI− staining, late apoptosis is defined by Annexin V+/PI+ staining, and necrosis is defined by Annexin V−/PI+ staining.

### 3.7. Cell Cycle Analyses

Cell cycle analyses were performed by using a Guava easyCyte flow cytometer (Merk-Millipore, MA, USA) and the DNA content was determined by propidium iodide (PI) method as previously described [42]. Briefly, A549, H1975, and HDF were treated with **APF-1** at 1.5 and 3 µM concentrations for 24 h. Cells were harvested, washed twice with cold PBS, and fixed with 70% ice-cold ethanol and kept at 20 °C for 2 h. The fixed cells were centrifugated at 4 °C and suspended in a master mix solution containing PBS, PI (40 µg/mL), and RNase (100 µg/mL) for 30 min at 37 °C in the dark. Samples were run through the Guava easyCyte flow cytometer set to internal cell cycle analysis protocol and 5000 cells per sample were measured. The percentage of cell cycle phases (G0/G1, S, and G2/M) was quantified using MultiCycle AV Cell Cycle in FCS Express 7 software (DeNovo Software, Pasadena, CA, USA). The experiment was performed three times in triplicate.

### 3.8. Western Blotting

Analysis of cell extracts (20 µg total proteins) was performed using SDS-PAGE 12% (Bio-Rad Laboratories, Hercules, CA, USA). Electrophoresis was carried out under reducing conditions, where uniformly negatively charged proteins move according to their mass. The proteins were then transferred to a PVDF membrane using the Trans-Blot Turbo instrument (Bio-Rad Laboratories, USA), after which the membrane was blocked with Every Blot Blocking Buffer (Bio-Rad Laboratories, USA). P21 protein levels were analyzed by Western blotting using a specific rabbit monoclonal primary antibody that specifically recognizes p21 protein (ab109199, Cambridge, MA, USA), and a secondary antibody anti-rabbit HRP (TA130023, OriGene Technologies, Rockville, MD 20850, USA) and detected by chemiluminescence using the Chemidoc device (Bio-Rad Laboratories, USA). Glyceraldehyde-3-phosphate dehydrogenase (GAPDH) was used as an internal control for protein loading, as a housekeeping gene that allows the normalization of the results obtained by Western blotting. GADPH protein levels were analyzed by Western blotting using a specific rabbit monoclonal primary antibody that specifically recognizes this protein (ARC0205, catalog number MA535235, Thermo Scientific, Waltham, MA, USA) and secondary antibody anti-rabbit HRP (TA130023, OriGene Technologies, Rockville, MD 20850, USA), also in this case detected by chemiluminescence with the Chemidoc instrument (Bio-Rad Laboratories, USA).

### 3.9. Statistical Analyses

Data are presented as means ± S.D. (standard deviations). Statistical comparison of differences among groups of data was carried out using one-way analysis of variance (ANOVA), followed by Tukey’s post hoc test using GraphPad Prism (San Diego, CA, USA). Values of *p* < 0.05 were considered statistically significant and values of *p* < 0.01 were considered highly significant.

## 4. Conclusions

Overall, a novel and promising aminophenoxy-functionalized flavone **APF-1** acting as selective anticancer drug for pulmonary cancer cells (NSCLC) has been identified. Structure–activity relationships underlined the importance of 4-aminophenoxy linked at 6′ of the A ring and of a phenoxy group on the B ring at C’3 position. The identified compound **APF-1** induced the accumulation of NSCLC cells in the G2/M phase through the up-regulation of p21^Waf1/Cip1^ expression. Further studies are essential to fully elucidate the molecular mechanisms responsible for the selectivity and efficacy of **APF-1**. Moreover, the optimization of **APF-1** structure could lead to enhancing its potential as a selective anticancer drug.

## Data Availability

Data are also reported in the Appendix A.

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
