# Peer review of "Identification of Flavone Derivative Displaying a 4′-Aminophenoxy Moiety as Potential Selective Anticancer Agent in NSCLC Tumor Cells"

_molecules, 2023, doi:10.3390/molecules28073239_

Round 1

Reviewer 1 Report

Identification of flavone derivative displaying a 4’-aminophenoxy moiety as potential selective anticancer agent in NSCLC 3 tumour cells

Giovanna Mobbili1, Brenda Romaldi2, Giulia Sabbatini1, Adolfo Amici2, Massimo Marcaccio3, Roberta Galeazzi1, 5 Emiliano Laudadio4, Tatiana Armeni2, Cristina Minnelli1*

Minor concerns to address:

All of the following are minor concerns that can/should be addressed prior to publication but do not detract from the scientific soundness or merit of the presented work.

Line 53 Figure 1 – What does A, B, and C stand for in Figure 1a.  The figure legend is not descriptive enough, it should be indicated in the figure legend that ABC refer to carbon rings?  Example: “Chemical structures of the flavone scaffold with carbon rings denoted A, B, or C”

Line 91 – Wording not clear.

Table 2 - The units for SI would not be in (uM).  Selectivity Index (SI) is not a measure of chemical concentration.

Line 136 - What is meant by “overall response rate of 30-50%”.  If this is in reference to a literature study, please provide a citation.

Figure 3d indicates that APF-1 is about 2X as effective on cancerous cells vs. HDF.  This suggests a much lower SI than indicated in Table 2.

Figure 4 – “(d)” in figure caption should be in front of “Immunoblot…”

Figure 4c – This is an abnormal profile for HDF cells in normal cell culture conditions.  It would be uncommon to have ~ 80% of cells in G1 phase unless they were arrested. Were these cells arrested in G1 phase? If not, can the author include another citation to indicate that this type of profile would be expected or has been seen in the literature.

Figure 4d&e - Please include the p21 data for HDF cells. While the data still suggests that p21 is up-regulated in response to APF-1 treatment, it would be more compelling to see if these results were consistent with the lack of cell cycle arrest in HDF cells presented in 4c. 

Author Response

Identification of flavone derivative displaying a 4’-aminophenoxy moiety as potential selective anticancer agent in NSCLC 3 tumour cells

Giovanna Mobbili1, Brenda Romaldi2, Giulia Sabbatini1, Adolfo Amici2, Massimo Marcaccio3, Roberta Galeazzi1, 5 Emiliano Laudadio4, Tatiana Armeni2, Cristina Minnelli1*

Minor concerns to address:

All of the following are minor concerns that can/should be addressed prior to publication but do not detract from the scientific soundness or merit of the presented work.

Dear reviewer,

Thank you very much for your careful analysis and valuable comments. Following your recommendations, we made some changes on our manuscripts.

Line 53 Figure 1 – What does A, B, and C stand for in Figure 1a.  The figure legend is not descriptive enough, it should be indicated in the figure legend that ABC refer to carbon rings?  Example: “Chemical structures of the flavone scaffold with carbon rings denoted A, B, or C”

Thank you for your suggestion. The legend is now corrected.

Line 91 – Wording not clear.

Thank you for your comments. We rephrase the sentence “To ascertain if the cytotoxicity of aminophenoxy flavone derivatives could be influenced to a different cell permeability, we experimentally determined the n-octanol-water partition coefficient (Ko/w) and the Lipinski’s Rule violations by SwissADME software” with “The cytotoxicity of APF derivatives could be influenced by their ability to cross the cellular membrane. Therefore, to assess the cell permeability, we experimentally determined the n-octanol-water partition coefficient (Ko/w) and the Lipinski’s Rule violations by SwissADME software.”

Table 2 - The units for SI would not be in (uM).  Selectivity Index (SI) is not a measure of chemical concentration.

Thank you for your comment. Sure, there is a writing error. The Table 1 is now modified by removing “µM” from SI column.

Line 136 - What is meant by “overall response rate of 30-50%”.  If this is in reference to a literature study, please provide a citation.

Thank you for your comment. I provided the references on the overall response rate of Doxorubicin and its related side effects in the text [19-23].

Figure 3d indicates that APF-1 is about 2X as effective on cancerous cells vs. HDF.  This suggests a much lower SI than indicated in Table 2.

Thank you for your comment. The differences found are linked to different APF-1 incubation time in the cell viability (Table 1) and apoptosis experiments (Fig. 3). The IC50 values, determined by MTT assay, was assessed after 72 h of treatment while the apoptosis as well as cell cycle analyses was determined after 24 h. We preferred to perform these experiments at 24 h to detect the early stages of apoptosis (line 151).  

Figure 4 – “(d)” in figure caption should be in front of “Immunoblot…”

Thank you for comment. The figure caption is now corrected.

Figure 4c – This is an abnormal profile for HDF cells in normal cell culture conditions.  It would be uncommon to have ~ 80% of cells in G1 phase unless they were arrested. Were these cells arrested in G1 phase? If not, can the author include another citation to indicate that this type of profile would be expected or has been seen in the literature.

Thank you for the suggestion. The HDF cell lines used in this work are primary human dermal fibroblasts from adults between passage 9-12. Cells were not artificially arrested in G1 phase (e.g., by serum deprivation). Treatment (including control) was done on cells with a confluence of about 55-60% and then they were evaluated after 24h. In this time frame the cells duplicated to about 78% confluence as shown by the cytogram. Of course, the tumor cells were treated to the same degree of confluence but show a different proliferative profile due to the specific reason that they are tumor cells. Some work has reported a similar trend to ours in the cell cycle profile of HDFs with a percentage of cells in G0/G1 phase of about 75% (DOI: 10.1002/advs.201901818; doi: 10.1186/2045-3701-4-24). These references have been included in the main text [25-26].

Figure 4d&e - Please include the p21 data for HDF cells. While the data still suggests that p21 is up-regulated in response to APF-1 treatment, it would be more compelling to see if these results were consistent with the lack of cell cycle arrest in HDF cells presented in 4c. 

This information could be very interesting, thank you for your suggestion. We performed therefore the WB analyses on HDF after APF-1 treatment and we did not observe a significative changes in the expression of p21. The Figure 4 was therefore modified by adding the blots of HDF and relative quantification. Thank you again for this valuable suggestion.  

Reviewer 2 Report

This is an interesting paper devoted to the synthesis and anticancer properties of five aminophenoxyflavone derivatives. I think these results may be of interest to Molecule readers. However, I recommend minor corrections in order to improve the description of the structural relationships of the presented compounds, as well as the synthesis pathway. Required fixes are listed below.

- Figure 1 is not clear. In addition, the use of a mixed labeling system; eg: "a)" - under Figure 1, while "A" - in the main text of "Introduction" and in the figure, one is bold, the other is not - similarly, the mix-match is in other figures.

- Scheme 1 is unclear - the presented synthesis route from Scheme 1 should be better related to Figure 2 (now it's implicit)

The study of the antiproliferative activity of all compounds was limited to NSCLC cells (H1975 and A549). Further evaluation of the effect on tumor cells (cell apoptosis) was performed only for APF1. However, the compound APF1 has also been obtained and described previously (Bioorg. Chem. 2022, 129, 106219, 449 doi:10.1016/j.bioorg.2022.106219).

This means that the newly synthesized compounds (APF2-APF5) have substituents/groups that induce less activity. Therefore, the presented description of the structure-activity relationship is too poor and too general. It also does not contain any information as to why the given structural modifications were introduced.

Author Response

This is an interesting paper devoted to the synthesis and anticancer properties of five aminophenoxyflavone derivatives. I think these results may be of interest to Molecule readers. However, I recommend minor corrections in order to improve the description of the structural relationships of the presented compounds, as well as the synthesis pathway. Required fixes are listed below.

Dear reviewer,

Thank you very much for your careful analysis and valuable comments. Following your recommendations, we made some changes on our manuscripts.

- Figure 1 is not clear. In addition, the use of a mixed labeling system; eg: "a)" - under Figure 1, while "A" - in the main text of "Introduction" and in the figure, one is bold, the other is not - similarly, the mix-match is in other figures.

Thank you for your comment. Now, the labeling system used is the same for the legend figures and the main text. In addition, we modified the legend of Figure 1 with “Figure 1. (a) Chemical structures of the flavone scaffold with carbon rings denoted A, B, C and (b) of known biologically active amino-functionalized flavones” to make it clearer.

- Scheme 1 is unclear - the presented synthesis route from Scheme 1 should be better related to Figure 2 (now it's implicit)

Thank you for your suggestion. We added a single figure containing both scheme 1 and Figure 2.

The study of the antiproliferative activity of all compounds was limited to NSCLC cells (H1975 and A549). Further evaluation of the effect on tumor cells (cell apoptosis) was performed only for APF1. However, the compound APF1 has also been obtained and described previously (Bioorg. Chem. 2022, 129, 106219, 449 doi:10.1016/j.bioorg.2022.106219).

This means that the newly synthesized compounds (APF2-APF5) have substituents/groups that induce less activity. Therefore, the presented description of the structure-activity relationship is too poor and too general. It also does not contain any information as to why the given structural modifications were introduced.

As you said, compound APF-1 was already synthesized in our previous study in which we searched a mutant-selective tyrosine kinase inhibitor against Epidermal Growth Factor Receptor (EGFR). In this context, APF-1 did not show efficacy against both mutants and wild type EGFR form. No data about the cytotoxicity of compound was reported. In the current work however, we studied the cytotoxicity of the APF-1 which showed low values of IC50 in the lung cancer cells and a higher selective index which led us to ascertain the influence of the 4’-aminophenoxy group position linked to the A ring of the flavone scaffold. To do this, we synthetized other 4 compounds in which the position of 4’-aminophenoxy group was different. The aim was therefore to study the influence of the 4’-aminophenoxy group position linked to the A ring of the flavone scaffold on cancer cytotoxicity. Between them, APF-1 remained the highest selective cytotoxic compound and showed higher ability to induce both apoptosis, cell cycle arrest and p21 downregulation. Compared to previous paper, the chemistry is not new, but the biological perspective is different giving information for further research into the design of flavone-based selective anticancer drugs. Therefore, we consider appropriate to publish now these promising results as a Communication and to reserve to a subsequent paper a complete structure-activity study that will require months of experiments.

The reason of the introduced structural modifications were added in the 2.1. subchapter.

Reviewer 3 Report

Dear all

the present work, chemistry work, has no novelty. The work is an extension of a previously published research DOI: 10.1016/j.bioorg.2022.106219 . However, the biological perspective is different and attractive. There is some minor remarks (see the attached file)

Author Response

Dear all

the present work, chemistry work, has no novelty. The work is an extension of a previously published research DOI: 10.1016/j.bioorg.2022.106219 . However, the biological perspective is different and attractive. There is some minor remarks (see the attached file).

Thank you for your comments. The manuscript is corrected following your suggestion and therefore the Scheme 1 and Figure 1 were joined together.

Reviewer 4 Report

The manuscript "Identification of flavone derivative displaying a 4’-aminophenoxy moiety as potential selective anticancer agent in NSCLC tumour cells" presents the synthesis and biological evaluation of five new flavone derivatives. The article is well written and organized, giving new insights on cell apoptosis, cell cycle arrest and involved pathways. Hence, it is suitable for publication after the following minor revisions:

- An in-depth SAR analysis is missing;

- At the end of line 108, add the range of concentrations used for the MTT assay;

- Few typo must be checked. For example, in scheme 1, Cs2CO3 is spelled wrong;

- Several literature references are missing. In particular, at the end of line 36 add examples of recent studies describing the "identification of novel anticancer drugs" such as: Eur J Med Chem. 2022 Dec 5;243:114744. doi: 10.1016/j.ejmech.2022.114744; PLoS One. 2022 Sep 23;17(9):e0272362. doi: 10.1371/journal.pone.0272362; RSC Adv. 2022 Nov 22;12(52):33525-33539. doi: 10.1039/d2ra06188kJ Med Chem. 2018 Feb 8;61(3):1375-1379. doi: 10.1021/acs.jmedchem.7b01388. Appropriare references are also missing at the end of line 148.

Author Response

The manuscript "Identification of flavone derivative displaying a 4’-aminophenoxy moiety as potential selective anticancer agent in NSCLC tumour cells" presents the synthesis and biological evaluation of five new flavone derivatives. The article is well written and organized, giving new insights on cell apoptosis, cell cycle arrest and involved pathways. Hence, it is suitable for publication after the following minor revisions:

Dear Reviewer,

thank you very much for your careful analyses and valuable comments. 

- An in-depth SAR analysis is missing;

Based on your suggestion, we added a figure that summarize the observed cytotoxicity results. The effect of the position of both 4-aminophenoxy and phenoxy moieties were discussed in the main text.

- At the end of line 108, add the range of concentrations used for the MTT assay;

Thank you for your suggestion, the concentrations of APF-1 tested was added at line 108.

- Few typo must be checked. For example, in scheme 1, Cs2CO3 is spelled wrong;

Thank you for your suggestion. The Scheme 1 was corrected. According to the right nomenclature, “4’-amminophenoxy”, when present, was replaced with “4-aminophenoxy”.

- Several literature references are missing. In particular, at the end of line 36 add examples of recent studies describing the "identification of novel anticancer drugs" such as: Eur J Med Chem. 2022 Dec 5;243:114744. doi: 10.1016/j.ejmech.2022.114744; PLoS One. 2022 Sep 23;17(9):e0272362. doi: 10.1371/journal.pone.0272362; RSC Adv. 2022 Nov 22;12(52):33525-33539. doi: 10.1039/d2ra06188k; J Med Chem. 2018 Feb 8;61(3):1375-1379. doi: 10.1021/acs.jmedchem.7b01388. Appropriare references are also missing at the end of line 148.

All references suggested were added [4-7]. In addition, we added also references at the end of line 148 [28, 29].